# Salivary Biomarkers in Periodontitis Post Scaling and Root Planing

**DOI:** 10.3390/jcm11237142

**Published:** 2022-12-01

**Authors:** K. Lakshmi Priya, Jaideep Mahendra, Little Mahendra, Anilkumar Kanakamedala, Khalaf F. Alsharif, Maryam H. Mugri, Saranya Varadarajan, Ahmed Alamoudi, Ali Abdel-Halim Abdel-Azim Hassan, Mrim M. Alnfiai, Khalid J. Alzahrani, Maha A. Bahammam, Hosam Ali Baeshen, Thodur Madapusi Balaji, Shilpa Bhandi

**Affiliations:** 1Department of Periodontics, Meenakshi Ammal Dental College and Hospital, Chennai 600095, Tamil Nadu, India; 2Department of Periodontics, Maktoum Bin Hamdan Dental University, Dubai 122002, United Arab Emirates; 3Department of Clinical Laboratory Sciences, College of Applied Medical Sciences, Taif University, P.O. Box 11099, Taif 21944, Saudi Arabia; 4Department of Maxillofacial Surgery and Diagnostic Sciences, College of Dentistry, Jazan University, Jazan 45412, Saudi Arabia; 5Department of Oral Pathology and Microbiology, Sri Venkateswara Dental College and Hospital, Chennai 600130, Tamil Nadu, India; 6Department of Oral Biology, Faculty of Dentistry, King Abdulaziz University, Jeddah 21589, Saudi Arabia; 7Department of Information Technology, College of Computers and Information Technology, Taif University, P.O. Box 11099, Taif 21944, Saudi Arabia; 8Department of Periodontology, Faculty of Dentistry, King Abdulaziz University, Jeddah 80209, Saudi Arabia; 9Executive Presidency of Academic Affairs, Saudi Commission for Health Specialties, Riyadh 11614, Saudi Arabia; 10Department of Orthodontics, Faculty of Dentistry, King Abdulaziz University, Jeddah 21589, Saudi Arabia; 11Department of Periodontology, Tagore Dental College and Hospital, Chennai 600127, Tamil Nadu, India; 12College of Dental Medicine, Roseman University of Health Sciences, South Jordan, UT 84095, USA

**Keywords:** salivary biomarkers, arginase, gingivitis, chronic periodontitis, root planing, scaling, uric acid

## Abstract

Objectives: This study was conducted to evaluate the levels of salivary uric acid and arginase in patients with periodontitis, generalized gingivitis, and in healthy individuals. Then, the effects of non-surgical periodontal therapy on levels of salivary arginase and uric acid were also investigated. Methods: A total of 60 subjects were divided into three groups based on periodontal health: group I comprised 20 healthy individuals; group II comprised 20 subjects who had generalized gingivitis; group III comprised 20 subjects who had generalized periodontitis. On day 0, the clinical examination of periodontal status was recorded, following which saliva samples were collected. Group II and group III subjects underwent non-surgical periodontal therapy. These patients were recalled on day 30 to collect saliva samples. The periodontal parameters were reassessed on day 90, and saliva samples were collected for analysis of salivary arginase and uric acid levels. Results: Group II and group III showed improvement in clinical parameters following non-surgical periodontal therapy on the 90th day. The MGI score, PPD, and CAL showed improvement. On day 0, at baseline, salivary arginase levels in group III and group II were higher than those in healthy subjects, whereas on day 0, salivary uric acid levels in group III and group II were lower than those in healthy subjects. Both on day 0 and day 90, the salivary arginase level showed a positive correlation with the periodontal parameters, whereas the salivary uric acid level was positively correlated with the periodontal parameters on day 90. Conclusion: the level of salivary arginase was a pro-inflammatory marker and a raised level of salivary uric acid was an anti-inflammatory marker following periodontal therapy, suggesting their pivotal role in assessing periodontal status and evaluation of treatment outcome.

## 1. Introduction

Periodontitis is an inflammatory disease of multifactorial origin, characterized by periods of exacerbation and remission [1]. The disease is initiated by the formation of plaque biofilm leading to a loss of equilibrium between microbial organisms and host response, resulting in disease progression. The inflammation of the surrounding periodontal tissues ultimately leads to alveolar bone and attachment loss [2]. The degree of tissue destruction has a log-linear relationship with the rise in microbial load. The main etiological factors for the disease include gram-negative anaerobic bacteria and a variety of facultative bacteria that reside in the subgingival biofilm [3].

Proper diagnosis and treatment modality has been recommended for periodontal and peri-implant diseases and conditions based on the recent classification reported by Caton et al. [4]. Various diagnostic methods are available for the early detection and diagnosis of periodontitis. They can be broadly divided into clinical and molecular methods. A biomarker or biologic marker is an objectively measured characteristic that provides an indicator of pharmacologic response, pathogenic processes, or normal biological processes after therapeutic intervention [5]. Analysis of biomarkers in fluids such as saliva, synovial fluid plasma, whole blood, cerebrospinal fluid, serum, semen, plasma, and urine have assisted in clinical diagnosis in medicine [6]. Among these sources, saliva is commonly used for early diagnosis and detection of biomarkers. Saliva collection is rapid, simple, and more importantly, painless, making it uncomplicated for screening purposes. Saliva contains systemically and locally derived markers related to periodontal disease, thus serving as a specific biomarker for the assessment of periodontitis [7].

Non-surgical periodontal therapy is the first recommended approach and gold standard to control periodontal infections. The primary objective of non-surgical therapy is to restore gingival health by removing the local factors that cause disease (plaque, calculus, and endotoxins). Hand and ultrasonic instruments are tools used in non-surgical therapy to drastically reduce the number of periodontopathogens in the oral environment, thus restoring gingival health [8]. The most relevant shortcoming of non-surgical therapy is the absence of a long-term effect due to an eventual bacterial recolonization after the therapy [8]. Accordingly, adjunctive treatments have been introduced in addition to SRP, e.g., the use of antibiotics, photodynamic therapy, administration of antioxidants, natural compounds, supplements (e.g., melatonin), and, in recent years, probiotic therapy to enhance the long-term effects of non-surgical therapy [9]. 

Salivary arginase and uric acid have been reported to play a significant role in periodontal disease [10,11]. Salivary arginase levels seem to be raised in chronic periodontitis, suggesting its role in the inflammatory process. Arginase is one of the five key enzymes of the urea cycle, which is essential for the metabolic conversion of ammonia into urea. Arginine, a semi-essential amino acid, mediates various physiologic processes, including urea and protein synthesis [10]. The increased levels of arginase reduce levels of nitric oxide, which is essential for destroying periodontal pathogens. This happens because both arginase and nitric oxide compete for the common substrate L-arginine. Through the urea cycle, arginase helps to produce polyamines, which are necessary for the survival of many oral bacteria. They provide nutritional support; hence, high arginase activity in saliva favors the growth of pathogens, leading to the progression of periodontal disease.

One of the major antioxidants present in saliva is uric acid, which is the most dominant antioxidant apart from glutathione, superoxide dismutase (SOD), albumin, and ascorbic acid. Uric acid is the end product of purine metabolism and it contributes to the antioxidant properties of saliva [11]. These biomarkers have both antioxidant and pro-oxidant properties in vitro by scavenging and producing reactive oxygen species and are known to have an impact during periodontal inflammation. Although studies in the past have shown the effects of these enzymes with regard to inflammation, very few studies have shown the effects of non-surgical therapy and periodontal status on levels of the above biomarkers. We hypothesized that decreased levels of salivary arginase and improved levels of uric acid post non-surgical therapy would have a positive impact on periodontal status. This study aims to quantify salivary arginase and uric acid levels in subjects with generalized gingivitis and periodontitis compared to those found in the healthy periodontium, and evaluate the effects of non-surgical periodontal therapy on salivary arginase and uric acid levels. 

## 2. Materials and Methods

Sixty subjects (30 males and 30 females) were recruited from the outpatient section of the Department of Periodontology, Meenakshi Ammal Dental College and Hospital, Chennai, India. Ethical approval was obtained from Institutional Review Board of MAHER—Deemed to be University, Chennai (MADC/IRB-XXV/2018/393) before commencement of the study. The power of the study was calculated using G-power software. The power was calculated for 60 subjects, with a minimum of 20 subjects required in each group, which came out to be 95% based on the following calculation.
**Calculation for power analysis:*****t*-tests—**Means: Difference between two independent means (two groups)**Analysis**: A priori: Compute required sample size**Input**:    Tail(s) = Two  Effect size d = 1.2041300  α err prob = 0.05  Power (1-β err prob) = 0.95  Allocation ratio N2/N1 = 1**Output**: Noncentrality parameter δ = 3.7113779  Critical t = 2.0280940  Df = 36 Sample size group 1 = 19  Sample size group 2 = 19  Total sample size = 38  Actual power = 0.9506005

Written informed consent was obtained from all participants of the study. The subjects were divided into three groups of twenty subjects based on their periodontal health. Group I consisted of 20 periodontally and systemically healthy subjects (Controls). Group II included subjects with generalized chronic gingivitis. Group III comprised patients with chronic periodontitis according to the current classification of periodontal and peri-implant diseases, 2017 (Figure 1). The inclusion criteria included subjects willing to participate in the study, within 30–65 years of age, with ≥10 natural teeth. 

Group I (control group) consisted of systemically healthy subjects with clinically healthy periodontium, probing pocket depth (PPD) ≤ 3 mm, without bone loss or attachment loss on radiographs [1]. 

Group II (experimental group) consisted of generalized chronic gingivitis subjects with PPD ≤ 3 mm having ≤ 10% bleeding on probing sites with clinical signs of gingival inflammation without signs of pseudo pockets, clinical attachment loss (CAL = 0), and radiographic evidence of bone loss.

Group III (experimental group) consisted of chronic periodontitis patients falling under Stage II/III, grade B category with interdental CAL detectable at ≥2 non-adjacent teeth or buccal or oral CAL ≥ 3 mm with pocketing > 3 mm and radiographic bone loss extending up to the middle third or apical third [1]. 

Individuals with a history of smoking, aggressive periodontitis, pregnant subjects, or subjects suffering from gout or systemic diseases that could affect periodontal status were excluded. Subjects with previous history of periodontal therapy in the last six months or were on medication affecting periodontal status were excluded from the study. 

### 2.1. Non-Surgical Periodontal Therapy

On day 0, after periodontal examination and collection of saliva samples for molecular analysis, non-surgical periodontal therapy was performed, which consisted of complete scaling and root planing along with oral hygiene instructions, on subjects in group II and group III by a trained periodontist (J.M. and K.L.P.). 

### 2.2. Parameters Assessed

A detailed dental and medical history of the subjects was noted at the initial visit, followed by a periodontal examination that included: the modified gingival index (MGI) [10], clinical attachment level (CAL), and probing pocket depth (PPD), which were measured using a William’s periodontal probe. Salivary arginase and uric acid levels were estimated at days 0, 30, and 90 (Figure 1). All parameters were assessed by calibrated investigators and taken as an average (K.L.P. and J.M.). 

### 2.3. Collection of Unstimulated Saliva Samples

Saliva sample collection was standardized according to the circadian rhythm of the subjects. The individuals were requested to clean their mouths completely by rinsing with distilled water before the saliva samples were collected. The patients were instructed to swallow any leftover saliva in their mouth following the oral rinse. A 5 mL sample of unstimulated saliva was obtained in the morning between the hours of 10 am to 11 am, two hours after the last meal. The participants were asked to keep their mouths closed and not cough up mucus while their saliva was collected. They were instructed to let their saliva pool to its utmost extent on the floor of their mouths and then expectorate into the collection vessel until the necessary amount was gathered. The samples were taken to the laboratory using a standard gel coolant pack maintaining a temperature between 2 and 4 °C for immediate testing.

### 2.4. Estimation of Salivary Arginase and Uric Acid Levels

Saliva samples were centrifuged for 15 min and their supernatant was stored at minus 20 ˚C until the assay was performed. The salivary arginase level was estimated using an arginase activity assay kit ^. All reagents were brought to room temperature before the procedure. Both the urea reagent and substrate buffer were freshly prepared before the assay and used within 2 h. The substrate buffer was prepared for each well of the reaction by mixing 4 μL of arginine buffer and 2 μL of Mn solution provided in the kit. The urea reagent was created by mixing 105 μL of both reagents A and B provided in the kit. An aliquot of 40 μL of arginase was transferred to each individual well. Two wells were reserved as blank wells. A 5 μL test sample (saliva) was added to the remaining wells containing the arginase. The plate was incubated for 15 min at 25 °C. An aliquot of 5 μL of substrate was added to all wells, except the blank well. A volume of 5 μL of distilled water was added to the blank well and incubated for 30 min at 25 °C. Following incubation, 200 μL of urea reagent was added to all wells to stop the arginase reaction. The plate was incubated at room temperature (37 °C). An ELISA plate reader (LABSERV) was used to measure the optical density (O.D.) at 430 nm. 

Uric acid analysis was performed using uric acid reagent (2 × 25 mL) and uric acid standard reagent (1.5 mL with a concentration of 8 mg/dL) **. The semi-auto analyzer was programmed to detect uric acid in the test samples at 510 nm. The kit contained ready-to-use components. The standard solution was pipetted and 0.5 μL was taken in Eppendorf tubes and incubated for 20 min at room temperature. The standard solution was aspirated through the analyzer and the concentration was measured to be 8 mg/dL, as provided by the uric acid estimation kit. An aliquot of 1.0 μL of uric acid reagent was taken in an Eppendorf tube and 0.5 μL of the test (saliva) sample was added. The contents were tapped, allowed to mix, and incubated for 10 min at room temperature. The working reagent along with the test sample was aspirated and the concentration of uric acid was estimated at 510 nm.

^ Sigma Aldrich—arginase activity assay kit (Sigma-Aldrich, St. Louis, MO, USA).

** Gen X uric acid estimation kit (Proton Biologicals India PVT LTD, Bangalore, Karnataka, India).

### 2.5. Statistical Analysis

Statistical analysis was performed using IBM SPSS software version 26.0 (IBM Corp., Armonk, NY, USA). The descriptive statistics were expressed using the mean and standard deviation. Shapiro-Wilk and Kolmogorov-Smirnov tests were used to assess the normality of the data. Based on the results obtained, the values followed a normal distribution. To compare mean values between groups, one-way ANOVA was applied, followed by Tukey’s HSD post hoc test for multiple pairwise comparisons. Karl Pearson correlations were calculated to assess the linear relationship between biomarkers and clinical variables. The probability value <0.05 was considered significant.

## 3. Results

Intra- and intergroup comparisons revealed that mean changes in clinical attachment level (CAL), probing pocket depth (PPD), and modified gingival index (MGI) from day 0 to day 90 were statistically significant (*p* = 0.000) for group II and group III (Table 1).

All three groups showed a significant difference (*p* = 0.000) in salivary arginase levels on days 0, 30, and 90. Salivary arginase levels were found to be higher in group III at all time points compared to those in group II. Although the intragroup comparison of salivary arginase levels was reduced at different time points, they did not show any statistically significant difference in mean change (Table 2).

Intergroup and intragroup assessments of salivary uric acid levels showed a statistically significant difference on day 0 among the groups. Intergroup comparisons of mean change in uric acid levels were statistically significant from day 0 to day 30. Similarly, intragroup comparisons also revealed statistically significant mean changes in uric acid levels from day 0 to day 30 and from day 0 to day 90 in group III (Table 2).

Karl Pearson correlation analysis was carried out to determine correlations between salivary arginase and uric acid levels with periodontal parameters on day 0 in all three groups. The results showed that both the levels of salivary arginase and uric acid were significantly correlated with MGI, PPD, and CAL on day 0 for all three groups (*p* = 0.000) (Table 3).

The salivary arginase levels were significantly correlated with the MGI, PPD, and CAL on day 90. However, the salivary uric acid level did not show any correlation with the clinical parameters on day 90 (Table 4).

## 4. Discussion

Periodontal disease is the sixth most prevalent disease globally, characterized by intermittent pain and destruction of tooth-supporting structures initiated by a complex microbial biofilm [12]. Saliva may prove to be an ideal diagnostic medium. Saliva contains a wide range of enzymes and molecules, including arginase, oxidized glutathione, uric acid, albumin, vitamin C, reduced glutathione, and SOD (superoxide dismutase), which work in concert to exert a defence against bacterial insult. Measuring the total enzyme status can reveal the severity of a disease, estimate the risk of diagnosing oral disease, and help monitor the host’s response to various oral treatments. Quantifying these biomarkers can also help identify periodontal risk factors [13,14].

Salivary arginase is considered to be essential in inflammation. Arginase is found mainly in the liver and salivary glands. This enzyme catalyzes the hydrolysis of L-arginine to urea and ornithine and forms one of the five key enzymes in the urea cycle. Arginase helps in the synthesis of polyamines in saliva, which are nutritionally important to the oral microbiota. Raised levels of arginase may lead to a decrease in NO (nitric oxide) synthesis and increased susceptibility to bacterial infection [15].

Uric acid is a major non-enzymatic enzyme present in saliva in healthy and periodontitis conditions [16]. Salivary uric acid levels may relate to plasma uric acid levels and can be used to assess enzyme status. Gozalez-Hernadez et al. reported uric acid as a major enzyme present in saliva, contributing to 70% of the total enzyme capacity [17]. It acts as a pro-oxidant or as an enzyme depending on the environment [18]. In hydrophilic conditions, it acts as an enzyme combating ROS. In a hydrophobic environment, it acts as a pro-oxidant. Raised levels of uric acid can induce situations associated with oxidative stress, such as obesity, hypertension, and cardiovascular diseases. Hyperuricemia is a diseased state that could also indicate a protective response against oxidative damage [18].

This study quantified salivary arginase and uric acid levels in patients with periodontitis and generalized gingivitis compared to those in periodontally healthy subjects. We also examined the impact of non-surgical periodontal therapy on salivary arginase and uric acid levels in these patients. 

Among the periodontal parameters assessed, the mean modified gingival index was reduced from day 0 to day 90 in group III and group II following scaling and root planing, which was statistically significant (Table 1, *p* = 0.000). These results are in accordance with earlier observations reported by Labao et al. and Kim et al. who found a reduction in inflammation following non-surgical therapy [19,20]. In intragroup and intergroup comparisons, we found the mean probing pocket depth (PPD) was significantly reduced from day 0 to 90 in group III and group II (*p* = 0.000). Van Der Weijden and Mesell et al. reported similar reductions in probing pocket depth following non-surgical therapy in chronic periodontitis subjects [21,22]. Similarly, in our study, there was an improvement in the mean clinical attachment level (CAL) from day 0 to 90 in group III and group II, which was also statistically significant (*p* = 0.000). These results corroborated the findings of Shah et al. and Schlagenhauf et al. who observed that non-surgical therapy resulted in a reduction of recession and clinical attachment gain [23,24]. Clinical attachment level is an important periodontal parameter that acts as an indicator of past tissue destruction, which helps to assess disease severity. Broadly consistent with previous research, we found that non-surgical therapy improved periodontal parameters in patients with gingivitis and periodontitis. 

The levels of salivary arginase and uric acid were assessed before and after non-surgical periodontal therapy on day 0, day 30, and day 90. L-arginine is a common substrate that is used by NOS to synthesize nitric oxide, which has antimicrobial properties against periodontal pathogens and host inflammatory cells [24,25]. In the present study, intergroup comparisons of salivary arginase levels on days 0, 30, and 90 in group III and group II showed statistical significance (Table 2, *p* = 0.000). Similar results were obtained by Castro et al. who observed increased salivary levels of arginase in periodontitis subjects and concluded that the enzymes can be used as a marker for periodontal inflammation [25]. However, intragroup comparisons of the mean change in salivary arginase levels in group II and group III did not show any statistical significance (*p* = 0.07 and *p* = 0.404, respectively). A possible explanation could be that both groups had periodontal inflammation and hence had higher levels of arginase. These findings suggest that salivary arginase activity in periodontitis, along with the arginine-nitric oxide pathway, may be involved in the disease process using the common substrate L-arginine and inhibiting nitric oxide production. Periodontal therapy may result in an improvement in levels of salivary arginase [25]. Our results suggest that arginase is associated with the inflammatory processes of periodontal disease and its activity is decreased concurrently with progression in clinical parameters, making it a potential inflammatory biomarker [11].

Various salivary antioxidants, such as superoxide dismutase, albumin, and ascorbic acid, have been used as markers for the diagnosis of periodontitis. Due to their high sensitivity and specificity, salivary uric acid and arginase can be potent and reliable markers for evaluating periodontal status [10]. We observed a positive correlation between periodontal markers (MGI, PPD, and CAL) and salivary arginase in groups II and III on day 0 and day 90 (Table 3 and Table 4). Similar results were obtained by Haririan et al. who found an improvement in clinical parameters with a reduction in salivary arginase levels following non-surgical periodontal treatment [26]. The subsequent reduction in levels of salivary arginase indicates that arginase levels are directly related to periodontal health status. 

The intergroup comparison of salivary uric acid levels on day 0 in all three groups was statistically significant (Table 2, *p* = 0.000). These results are similar to those reported by Rizal and Vega, Pattanshetti et al., and Uppin et al. who found reduced levels of salivary uric acid in periodontitis patients compared to those in healthy subjects [11,27,28]. The difference in uric acid levels among the groups may be attributed to bacterial variability in gingivitis and periodontitis. We did not observe any significant difference in uric acid levels in group II and group III on day 30 and day 90 following non-surgical therapy. 

The intragroup comparison of salivary uric acid levels in group II at different time points did not show any significant difference. However, in group III, the differences were statistically significant (Table 2, *p* = 0.000). These results are likely related to the inflammatory burden and oxidative stress in periodontitis being markedly higher than in gingivitis. Hence, non-surgical periodontal therapy that reduces inflammation could have a profound impact on salivary uric acid levels [29]. Uric acid levels tend to rise after non-surgical periodontal therapy, further facilitating healing of the tissues [28]. Nominal levels of oxidative stress trigger the action of protective enzymatic mechanisms, resulting in maintenance of the structural integrity of the periodontium. Non-surgical periodontal therapy results in a significant decrease in the bacterial load and oxidative stress in the periodontal tissues. The elimination of oxidative stress results in variations in the levels of uric acid, suggesting potential biomarkers for oral health [30]. Similar results were obtained by Sayar et al. and Baz et al. in gingivitis patients following non-surgical therapy [31,32]. An increase in salivary uric acid levels in periodontitis patients was attributed to a decrease in the abundance of free radicals following periodontal treatment [15].

Salivary uric acid levels showed a positive correlation with periodontal parameters (MGI, PPD, and CAL) on day 0, which was statistically significant (Table 3, *p* = 0.000). The improvement in periodontal status through non-surgical periodontal therapy increased salivary uric acid levels. Our result reflects those of Sayar et al. who observed a rise in uric acid levels with improvement in the periodontal status [31]. Bacterial invasion can cause a loss of balance between ROS and antioxidant defense, which contributes to the pathogenesis of periodontal disease. Similarly, Baz et al. found that the excessive amount of free radicals produced by periodontal inflammation exhausted the defensive capabilities of this enzyme, leading to tissue damage [32]. A decrease in free radical abundance can be achieved by eliminating inflammation through non-surgical therapy such as scaling and root planing, thereby neutralizing and restoring oxidative homeostasis and leading to improvement in the periodontal status. However, on the 90th day, salivary uric acid levels did not show any positive correlation with periodontal parameters (Table 4). The mouth is a heterogeneous environment for the resident microbiota but offers several distinct habitats for microbial colonization. These oral habitats form a highly heterogeneous ecological system and support the growth of significantly different microbial communities. The alternating warm and moist environment in the oral cavity suits the growth of many microorganisms and offers host-derived nutrients, such as saliva proteins, glycoproteins, and gingival crevicular fluid (GCF), among which the current study highlighted the effects of salivary uric acid and arginase on periodontal disease progression. In addition, an unbalanced oral microbiome could be detrimental not only to oral health but also have an effect on general health. Thus, balancing the host oral microbiota from a dysbiotic to a symbiotic environment is needed to achieve cessation of disease progression, which could be achieved by evolving adjunct pro- and prebiotic therapy [33]. 

The results of our study indicate the role and importance of salivary arginase in periodontal disease and highlight the relationship between periodontal inflammation and enzyme levels. Our results demonstrated that non-surgical periodontal therapy is capable of eliminating the inflammatory burden and producing a profound impact on levels of uric acid in groups II and III from day 0 to 90 after non-surgical periodontal treatment. Hence, salivary uric acid and arginase can be utilized as sensitive and specific biomarkers for the early detection of periodontal inflammation and tissue destruction, since they exert a definitive role in periodontal disease destruction and are widely distributed in saliva.

One limitation of our study was that only salivary enzymes were measured. In future, serum enzymatic levels can also be assessed and compared with salivary enzyme levels in order to authenticate the results. Further research can validate our findings by assessing the gene expression of arginase using RT-PCR. Another limitation of our study was the use of saliva as a diagnostic tool for the detection of biomarkers as it is prone to contamination and concentration variability of proteins and enzymes. GCF samples, as a more specific diagnostic tool, could be implemented for collecting oral samples. Further, future studies exploring the effects of non-surgical periodontal therapy on arginase and uric acid receptors and inhibitors in the gingival crevicular fluid would more specifically clarify the patho-immunogenic link between these biomarkers and periodontal disease. 

## 5. Conclusions

The study showed decreased salivary arginase levels in patients with gingivitis and periodontitis following the non-surgical periodontal therapy (NSPT). Salivary arginase levels were also found to be positively correlated with periodontal parameters. On the other hand, salivary uric acid levels showed improvement following NSPT in these patients. Our findings indicate that levels of salivary arginase and uric acid can be used as potential biomarkers in the early detection of periodontal inflammation and indicators following periodontal therapy. In future, these biomarkers could help provide a non-invasive, chair-side diagnosis in the era of advancing diagnostic technology in order to provide point-of-care diagnosis, screening, and monitoring of treatment efficacy.

## Figures and Tables

**Figure 1 jcm-11-07142-f001:**
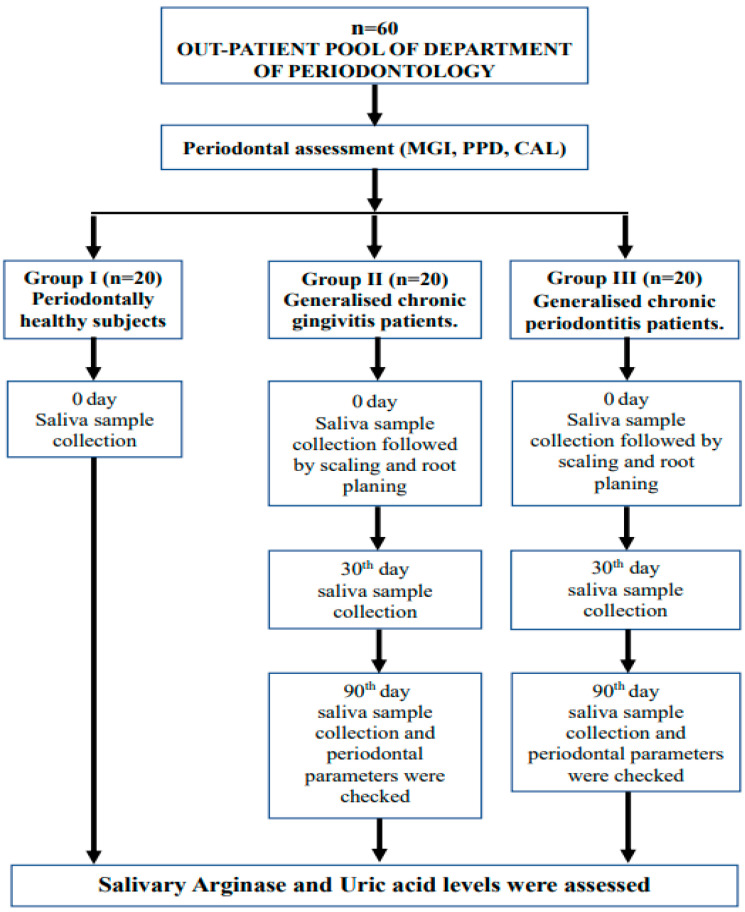
Flowchart of the study design.

**Table 1 jcm-11-07142-t001:** Comparison of standard deviation, mean, and significant differences for periodontal parameters (MGI, PPD, CAL) within and between groups at different time intervals.

Periodontal Parameters	Time Interval (Days)	Group I (Mean ± SD)	Group II (Mean ± SD)	Group III (Mean ± SD)	*p*-Value
Modified gingival index	0	0.42 ± 0.20	1.17 ± 0.20	2.52 ± 0.21	0.000 *
	90	-	0.30 ± 0.13	1.66 ± 0.28	0.000 *
	Mean change 0 to 90	-	0.86 ± 0.20	0.86 ± 0.27	0.984 ^NS^
	*p*-value	-	0.000 *	0.000 *	
Probing pocket depth (mm)	0	1.36 ± 0.27	1.48 ± 0.14	5.10 ± 0.26	0.000 *
	90	-	1.18 ± 0.07	2.29 ± 0.48	0.000 *
	Mean change 0 to 90	-	0.29 ± 0.15	2.80 ± 0.52	0.000 *
	*p*-value	-	0.000 *	0.000 *	
Clinical attachment level (mm)	0	1.35 ± 0.27	1.48 ± 0.14	5.47 ± 0.23	0.000 *
	90	-	1.18 ± 0.07	2.41 ± 0.62	0.000 *
	Mean change 0 to 90	-	0.29 ± 0.15	3.05 ± 0.60	0.000 *
	*p*-value	-	0.000 *	0.000 *	

*—Statistically significant; ^NS^—Statistically not significant. Level of statistical significance *p* ˂ 0.05.

**Table 2 jcm-11-07142-t002:** Comparison of mean, standard deviation, and significant differences for salivary arginase and uric acid levels within and between groups at different time intervals.

Salivary Parameters	Time Intervals (Days)	Group I	Group II	Group III	*p*-Value
Salivary arginase levels Units/L	0	3.21 ± 4.15	9.20 ± 5.93	14.31 ± 6.02	0.000 *
	30	-	7.74 ± 1.30	13.70 ± 1.37	0.003 *
	90	-	5.43 ± 1.17	11.18 ± 1.46	0.004 *
	*p*-value	-	0.076 ^NS^	0.404 ^NS^	-
	Mean change 0 to 30th day	-	1.46 ± 8.32	0.60 ± 7.23	0.731 ^NS^
	*p*-value	-	1.000 ^NS^	1.00 0 ^NS^	
	Mean change 0 to 90th day	-	3.74 ± 6.75	3.13 ± 10.43	0.818 ^NS^
	*p*-value	-	0.066 ^NS^	0.586 ^NS^	
	Mean change 30th to 90th day	-	2.31 ± 8.32	2.52 ± 8.71	0.938 ^NS^
	*p*-value		0.687 ^NS^	0.632 ^NS^	
Salivary uric acid levels mg/dL	0	21.49 ± 10.01	9.73 ± 7.56	5.64 ± 4.32	0.000 *
	30	-	10.21 ± 11.19	16.30 ± 13.91	0.136 ^NS^
	90	-	16.35 ± 12.10	18.54 ± 10.01	0.537 ^NS^
	*p*-value	-	0.127 ^NS^	0.000 *	-
	Mean change 0 to 30th day	-	0.48 ± 12.27	10.66 ± 13.68	0.018 *
	*p*-value	-	1.000 ^NS^	0.007 *	
	Mean change 0 to 90th day	-	−6.62 ± 13.50	−12.90 ± 10.22	0.106 ^NS^
	*p*-value	-	0.123 ^NS^	0.000 *	
	Mean change 30th to 90th day	-	−6.13 ± 16.09	−2.23 ± 16.34	0.452 ^NS^
	*p*-value	-	0.31 ^NS^	1.000 ^NS^	

*—Statistically significant; ^NS^—Statistically not significant; Level of statistical significance *p* ˂ 0.05.

**Table 3 jcm-11-07142-t003:** Karl Pearson correlation analysis of salivary arginase and uric acid levels and all periodontal parameters on day 0for all three groups (I, II, and III).

Periodontal Parameters		Salivary Arginase (units/L) (N = 60)	Salivary Uric Acid (mg/dL) (N = 60)
Modified gingival index	CORRELATION	0.625	−0.568
	*p*-value	0.000 *	0.000 *
Probing pocket depth (mm)	CORRELATION	0.556	−0.458
	*p*-value	0.000 *	0.000 *
Clinical attachment level (mm)	CORRELATION	0.555	−0.461
	*p*-value	0.000 *	0.000 *

*—Statistically significant; Level of statistical significance *p* ˂ 0.05.

**Table 4 jcm-11-07142-t004:** Karl Pearson correlation analysis of salivary arginase and uric acid levels and all periodontal parameters on the 90th day for groups II and III.

Periodontal Parameters		Salivary Arginase (units/L) (n = 40)	Salivary Uric Acid (mg/dL) (n = 40)
Modified gingival index	CORRELATION	0.426	0.124
	*p*-value	0.006 *	0.447 ^NS^
Probing pocket depth (mm)	CORRELATION	0.343	0.100
	*p*-value	0.03 *	0.538 ^NS^
Clinical attachment level (mm)	CORRELATION	0.316	0.090
	*p*-value	0.047 *	0.583 ^NS^

*—Statistically significant; ^NS^—Statistically not significant; Level of statistical significance *p* ˂ 0.05.

## Data Availability

Not applicable.

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
