# Peer review of "Salivary Biomarkers in Periodontitis Post Scaling and Root Planing"

_jcm, 2022, doi:10.3390/jcm11237142_

Round 1

Reviewer 1 Report

Manuscript of considerable interest, with the focus on bacterial management during periodontal treatment.

A major revision is needed before evaluating the possibility of the relationship.

In my opinion, the authors are too many

Well described abstract.

Few keywords, to add specific ones.

Introduction: the definition of periodontal disease has been correctly defined based on the new classification, but all the adjuvants of the minimally invasive treatment are missing, such as: ozone, laser, ozonated water, probiotics, paraprobiotics, post biotics, as already studied by the research group of prof. Scribant.

Well described materials and methods, but the calculation of the sample size is missing.

Results to be highlighted more statistically significant data.

Discussion add proactive action to maintain a balanced microbiota as future goals.

Conclusions, reformulate them according to the changes in the text

Bibliography: add required references

Author Response

REVIEWER 1

A major revision is needed before evaluating the possibility of the relationship.

In my opinion, the authors are too many

COMMENT 1:Well described abstract.

ANSWER: Thanks for the reviewer’s comments.

COMMENT 2: Few keywords, to add specific ones.

ANSWER: The key words have been modified as per reviewer’s suggestion and highlighted.

COMMENT 3: Introduction: the definition of periodontal disease has been correctly defined based on the new classification, but all the adjuvants of the minimally invasive treatment are missing, such as: ozone, laser, ozonated water, probiotics, paraprobiotics, post biotics, as already studied by the research group of prof. Scribant.

ANSWER: The reviewer’s comments are welcome. The adjuvants of the minimally invasive treatment has been added and highlighted as per reviewer’s suggestions. 

COMMENT 4: Well described materials and methods, but the calculation of the sample size is missing.

ANSWER: The calculation of sample size is being added and highlighted as per reviewer’s suggestions.

COMMENT 5:Results to be highlighted more statistically significant data.

ANSWER: The results have been modified by highlighting the statistically significant data as per reviewer’s suggestions.

COMMENT 6:Discussion add proactive action to maintain a balanced microbiota as future goals.

ANSWER: The proactive action to maintain a balanced microbiota as future goals has been added and highlighted as per reviewer’s suggestions.

COMMENT 7:Conclusions, reformulate them according to the changes in the text

ANSWER: The conclusion has been reformulated as per changes in the text according to reviewer’s suggestion.

COMMENT 8:Bibliography: add required references

ANSWER: The references have been added and highlighted as per reviewer’s suggestions.

Reviewer 2 Report

The topic of the present paper is interesting and it discusses certain salivary biomarkers present in inflamatory periodontal disease. Howerver there are several concerns regarding the manuscript :

1. references are too old; please keep references older than 10 years to a minimum and add more relevant papers published within the last 5 years.

2. please better justify why the authors chose just arginase and uric acid for this particular study

3. regarding statistics: please explain why the following statistical methods where chosen for the present data: Shapiro-Wilk and Kolmogorov-Smirnov tests

4. one main concern is the number of subjects present in the study; it is too low

5. another major issue is the novelty of the study: while the ideea in itself has merit , it is not original and has previously been published in other studies. In other words: what is the novelty of the this study. 

Author Response

REVIEWER 2

The topic of the present paper is interesting and it discusses certain salivary biomarkers present in inflamatory periodontal disease. Howerver there are several concerns regarding the manuscript :

  1. COMMENT 1: references are too old; please keep references older than 10 years to a minimum and add more relevant papers published within the last 5 years.

ANSWER: The recent references  has been highlighted as per reviewer’s suggestion.

  1. COMMENT 2: please better justify why the authors chose just arginase and uric acid for this particular study

ANSWER: The novelty of the biomarkers has been added at the end of the discussion part and has been highlighted as per reviewer’s suggestions

COMMENT 3: regarding statistics: please explain why the following statistical methods where chosen for the present data: Shapiro-Wilk and Kolmogorov-Smirnov tests

ANSWER: The statistical analysis has been modified and has been made more specific correlating with results and tables.

  1. COMMENT 4: one main concern is the number of subjects present in the study; it is too low

ANSWER: We thank the reviewer’s suggestions. The subjects were chosen based on the sample size calculation. Moreover, since the current study was a follow up study, we documented many drop outs. Finally only 20 samples in each group were considered. However, this study needs more number of sample size to be incorporated which would be carried out in our future research work.

  1. COMMENT 5: another major issue is the novelty of the study: while the idea in itself has merit, it is not original and has previously been published in other studies. In other words: what is the novelty of the this study

ANSWER: The novelty of the study has been addressed at the end part of discussion as per reviewer’s suggestion. 

Round 2

Reviewer 1 Report

The manuscript has been properly revised

Author Response

Reviewer 1 - Round 2:

The manuscript has been properly revised.

Reviewer 2 - Round 2:

The manuscript has not been significantly improved following first round of review

- my original comments and concerns still stand and have not been thoroughly addressed.

ANSWER: We regret that the corrections made were not up to the reviewer’s suggestion. We have worked on the article again and have improved the answers to the comments.

Editor notes:

Dear Authors,
Concerns highlighted by Reviewer 2 during the first round of revisions have been not taken care properly. As your article is of a certain interest, we have decided to give you a further opportunity to improve your work. Please improve your corrections and write a new response letter to the reviewer.

The topic of the present paper is interesting and it discusses certain salivary biomarkers present in inflammatory periodontal disease. However there are several concerns regarding the manuscript:

Authors reply to the comments:

The article has been revised and the answers of the comments of Reviewer 2 has been addressed a per to our satisfaction. We thank the reviewers for their valuable suggestions. 

REVIEWER 2

The topic of the present paper is interesting and it discusses certain salivary biomarkers present in inflamatory periodontal disease. However there are several concerns regarding the manuscript:

1.COMMENT 1: references are too old; please keep references older than 10 years to a minimum and add more relevant papers published within the last 5 years.

ANSWER: The recent references has been added as per reviewer’s suggestion.

  1. COMMENT 2: Please better justify why the authors chose just arginase and uric acid for this particular study

ANSWER: The novelty of the biomarkers has been added at the end of the introduction part and has been highlighted as per reviewer’s suggestions.

COMMENT 3: Regarding statistics: please explain why the following statistical methods where chosen for the present data: Shapiro-Wilk and Kolmogorov-Smirnov tests

ANSWER: The explanation has been given regarding the statistical analysis and the same has been modified and made more specific based on the results and tables.

  1. COMMENT 4: one main concern is the number of subjects present in the study; it is too low

ANSWER: The subjects were chosen based on the sample size calculation. Moreover, since the current study was a follow up study, we documented many drop outs. In future more sample size would be incorporated for the better authentication.

  1. COMMENT 5: another major issue is the novelty of the study: while the idea in itself has merit, it is not original and has previously been published in other studies. In other words: what is the novelty of this study

ANSWER: Though the salivary enzymes such as arginase and uric acid has been discussed in the past but with a very limited literature. Moreover, the effect of non-surgical therapy has not been addressed much as one of the periodontal interventions in many of the past studies. This study intends to explore the effect of NSPT on the salivary enzyme levels. The novelty of the study has been discussed and addressed at the end part of introduction as per reviewer’s suggestion. 

Reviewer 2 Report

- manuscript has not been significantly improved following first round of review

- my original comments and concerns still stand and have not been thoroughly addressed 

Author Response

(The authors gave the same response as above.)
